## [Peer Review file · Nature Structural & Molecular Biology]

Global re-organisation of genome architecture at the transition to gametogenesis

Corresponding Author: Professor Petra Hajkova

Version 0:

Decision Letter:

Our ref: NSMB-A51762-T

14th Nov 2025

Dear Professor Hajkova,

I am writing on behalf of my chief editor, Dr. Perdigoto, the handling editor of your manuscript "Global re-organisation of genome architecture at the transition to gametogenesis" (NSMB-A51762-T), who is currently out of office but did not want her absence to delay an editorial decision and thus assigned me to send one once all reports were in and we had editorially discussed them. Your manuscript has now been seen by the original referees and their comments are below. Please note that Reviewers #1 and #4 co-reviewed the manuscript. The referees find that the paper has improved in revision, and therefore we are happy to accept it in principle in Nature Structural & Molecular Biology, pending minor revisions to satisfy the referees' final requests and to comply with our editorial and formatting guidelines.

We are now performing detailed checks on your paper and will send you a checklist detailing our editorial and formatting requirements in about 2 weeks. Please do not upload the final materials and make any revisions until you receive this additional information from us.

Sincerely,

Dimitris Typas
Senior Editor
Nature Structural & Molecular Biology
ORCID: 0000-0002-8737-1319

on behalf of

Carolina Perdigoto, PhD
Chief Editor
Nature Structural & Molecular Biology

Reviewer #1 (Remarks to the Author):

The authors have addressed most of my previous comments and concerns, and the manuscript has been substantially improved. The additional data further support the main conclusions. I do not have further comments regarding the data. However, a few technical limitations remain, which can be addressed through textual revisions. In particular, some statements remain somewhat overstated and could be toned down or rephrased, as noted below.

Major comments:

1. The Hi-C analysis of PGCs was performed at only one developmental time point (E13.5) and compared solely with gonadal somatic cells from the same stage. Therefore, conclusions such as "disruption of TADs," "loss of detectable loops," and "reduced active-active compartment interactions" are relative to somatic cells. The manuscript should avoid language that implies developmental transitions or dynamic changes within germ cells. For example, line 10 of the abstract states

“disruption of TAD, loss of detectable loops, and reduced compartment interactions,” but it is unclear what these features are being compared to. This phrasing is potentially misleading and should be clarified throughout the manuscript by explicitly stating that these observations are relative to somatic cells.

2. The observed differences in 3D genome organization between in vivo PGCs and PGCLCs are interesting; however, the data do not establish a causal link with meiotic competency. It is acceptable to discuss this as a possible explanation in the discussion, but stating it explicitly in the abstract constitutes an overinterpretation. The authors should remove this statement from the abstract and adjust related descriptions in the main text to avoid implying causality.

3. Since the distinction in 3D genome architecture between in vivo PGCs and PGCLCs is a central finding, it would help readers to visualize these differences in a schematic model. Consider adding a panel to Figure 8 or Extended Data Figure 10 summarizing the major contrasts between PGCs and PGCLCs.

Minor comments:

1. Figures 1c and 1d should be swapped, as Figure 1d is referred to first in the main text.
2. Line 99 – A brief explanation of ATRX should be included.
3. Figures 3h and 3i appear before Figure 3g in the main text.
4. Extended Data Figure 4c appears before Figure 4b in the main text.
5. Figure 4c is not mentioned in the main text.
6. Figure 7c is not mentioned in the main text.

Reviewer #2 (Remarks to the Author):

Regarding our major concerns.

The authors now provide a Western blot quantification of LaminB1, LBR and CTCF. Results confirm the IF. It is not very clear for CTCF but a small decrease is visible in the quantification.

They performed IFs for CENPA and CTCF to assess the continuation of the configuration later in spermatogenesis. The peripheral signal for CENPA is maintained in undifferentiated spermatogonia, which is nice to show and agrees with previous data. For spermatocytes, the legend may be misleading as they refer to "telomere clustering" while using a centromeric antibody. It may be more appropriate to refer to "bouquet formation". The CTCF IFs are also convincing.

Regarding the minor comments, they have addressed them all.

In conclusion, we are already convinced of the quality and impact of the manuscript in the former submission. In light of these new additions, we further support publication.

Reviewer #3 (Remarks to the Author):

I reviewed the original submission of this manuscript. The authors have performed a thorough set of revision experiments, and they have addressed all of my comments. I support publication of the revised manuscript.

Reviewer #4 (Remarks to the Author):

I co-reviewed this manuscript with one of the reviewers who provided the listed reports. This is part of the Nature Structural & Molecular Biology initiative to facilitate training in peer review and to provide appropriate recognition for Early Career Researchers who co-review manuscripts.

Version 1:

Decision Letter:

12th Jan 2026

Dear Professor Hajkova,

We are now happy to accept your revised paper "Global re-organisation of genome architecture at the transition to gametogenesis" for publication as an Article in Nature Structural & Molecular Biology.

Your paper will be published online soon after we receive proof corrections and will appear in print in the next available issue. You can find out your date of online publication by contacting the production team shortly after sending your proof corrections.

Authors may need to take specific actions to achieve compliance with funder and institutional open access mandates. If your research is supported by a funder that requires immediate open access (e.g. according to <https://www.springernature.com/gp/open-science/plan-s-compliance> Plan S principles or the <https://www.springernature.com/gp/open-science/us-federal-agency-compliance> NIH public access policy) then you should select the gold OA route, and we will direct you to the compliant route where possible. Because authors warrant under our subscription licensing terms that they haven't committed to licensing any version of their article under a licence inconsistent with the terms of our agreement – including the applicable embargo period – publication under the subscription model isn't suitable for authors whose funders require no embargo.

Sincerely,

Dimitris Typas
Senior Editor
Nature Structural & Molecular Biology
ORCID: 0000-0002-8737-1319

Response to the Reviewers' comments:

Reviewer #1 (Remarks to the Author):

The authors have addressed most of my previous comments and concerns, and the manuscript has been substantially improved. The additional data further support the main conclusions. I do not have further comments regarding the data. However, a few technical limitations remain, which can be addressed through textual revisions. In particular, some statements remain somewhat overstated and could be toned down or rephrased, as noted below.

Major comments:

1. The Hi-C analysis of PGCs was performed at only one developmental time point (E13.5) and compared solely with gonadal somatic cells from the same stage. Therefore, conclusions such as “disruption of TADs,” “loss of detectable loops,” and “reduced active-active compartment interactions” are relative to somatic cells. The manuscript should avoid language that implies developmental transitions or dynamic changes within germ cells. For example, line 10 of the abstract states “disruption of TAD, loss of detectable loops, and reduced compartment interactions,” but it is unclear what these features are being compared to. This phrasing is potentially misleading and should be clarified throughout the manuscript by explicitly stating that these observations are relative to somatic cells. Abstract has been revised as requested.

2. The observed differences in 3D genome organization between in vivo PGCs and PGCLCs are interesting; however, the data do not establish a causal link with meiotic competency. It is acceptable to discuss this as a possible explanation in the discussion, but stating it explicitly in the abstract constitutes an overinterpretation. The authors should remove this statement from the abstract and adjust related descriptions in the main text to avoid implying causality.

We have removed this sentence from the abstract as requested. Apart from the abstract, potential links to meiotic competence are only mentioned in the discussion, which this reviewer found acceptable.

3. Since the distinction in 3D genome architecture between in vivo PGCs and PGCLCs is a central finding, it would help readers to visualize these differences in a schematic model. Consider adding a panel to Figure 8 or Extended Data Figure 10 summarizing the major contrasts between PGCs and PGCLCs.

Our manuscript focuses on spatial genome organization observed in germ cells in developing embryos. PGCLCs represent a heterogeneous population, with variable developmental competence. While it is important to know that in vitro PGCLCs do not fully recapitulate the observation in vivo. We prefer not to dilute the attention from the key physiological findings.

Minor comments:

1. Figures 1c and 1d should be swapped, as Figure 1d is referred to first in the main text. revised as requested.

2. Line 99 – A brief explanation of ATRX should be included. revised as requested.

3. Figures 3h and 3i appear before Figure 3g in the main text. revised as requested.

4. Extended Data Figure 4c appears before Figure 4b in the main text. We are not certain about this concern.
5. Figure 4c is not mentioned in the main text. Figure 4c is mentioned originally.
6. Figure 7c is not mentioned in the main text. revised as requested.

Reviewer #2 (Remarks to the Author):

Regarding our major concerns.

The authors now provide a Western blot quantification of LaminB1, LBR and CTCF. Results confirm the IF. It is not very clear for CTCF but a small decrease is visible in the quantification.

They performed IFs for CENPA and CTCF to assess the continuation of the configuration later in spermatogenesis. The peripheral signal for CENPA is maintained in undifferentiated spermatogonia, which is nice to show and agrees with previous data. For spermatocytes, the legend may be misleading as they refer to "telomere clustering" while using a centromeric antibody. It may be more appropriate to refer to "bouquet formation". The CTCF IFs are also convincing.

This has been changed as the reviewer suggested.

Regarding the minor comments, they have addressed them all.

In conclusion, we are already convinced of the quality and impact of the manuscript in the former submission. In light of these new additions, we further support publication.

Reviewer #3 (Remarks to the Author):

I reviewed the original submission of this manuscript. The authors have performed a thorough set of revision experiments, and they have addressed all of my comments. I support publication of the revised manuscript.

Reviewer #4 (Remarks to the Author):

I co-reviewed this manuscript with one of the reviewers who provided the listed reports. This is part of the Nature Structural & Molecular Biology initiative to facilitate training in peer review and to provide appropriate recognition for Early Career Researchers who co-review manuscripts.